# Personality Traits or Genetic Determinants—Which Strongly Influences E-Cigarette Users?

**DOI:** 10.3390/ijerph17010365

**Published:** 2020-01-05

**Authors:** Anna Grzywacz, Aleksandra Suchanecka, Jolanta Chmielowiec, Krzysztof Chmielowiec, Kamila Szumilas, Jolanta Masiak, Łukasz Balwicki, Monika Michałowska-Sawczyn, Grzegorz Trybek

**Affiliations:** 1Independent Laboratory of Health Promotion of the Pomeranian Medical University in Szczecin, 11 Chlapowskiego St., 70-204 Szczecin, Poland; o.suchanecka@gmail.com; 2Department of Hygiene and Epidemiology, Collegium Medicum, University of Zielona Góra, 28 Zyty St., 65-046 Zielona Góra, Poland; chmiele1@o2.pl (J.C.); chmiele@vp.pl (K.C.); 3Department of Physiology, Pomeranian Medical University in Szczecin, Powstańców Wlkp.72, 70-111 Szczecin, Poland; kamila.szumilas@pum.edu.pl; 4Neurophysiological Independent Unit, Department of Psychiatry, Medical University of Lublin, 20-093 Lublin, Poland; jolantamasiak@wp.pl; 5Department of Public Health and Social Medicine, Medical University of Gdansk, 42A Zwyciestwa St., 80-210 Gdansk, Poland; balwicki@gumed.edu.pl; 6Faculty of Physical Culture, Gdańsk University of Physical Education and Sport, 80-853 Gdańsk, Poland; monikamichalowska@op.pl; 7Department of Oral Surgery, Pomeranian Medical University in Szczecin, 72 Powstańców Wlkp. St., 70-111 Szczecin, Poland; g.trybek@gmail.com

**Keywords:** nicotine dependency, e-cigarettes, personality, dopamine

## Abstract

Presently, a growing popularity of electronic cigarettes may be observed. Used as a means of obtaining nicotine they allow to substitute traditional cigarettes. The origins of substance use disorders are conditioned by dopaminergic signaling which influences motivational processes being elementary factors conditioning the process of learning and exhibiting goal-directed behaviors. The study concentrated on analysis of three polymorphisms located in the dopamine receptor 2 (DRD2) gene—rs1076560, rs1799732 and rs1079597 using the PCR method, personality traits determined with the Big Five Questionnaire, and anxiety measured with the State Trait Anxiety Inventory. The study was conducted on a group of 394 volunteers, consisting e-cigarette users (*n* = 144) and controls (*n* = 250). Compared to the controls the case group subjects achieved significantly higher scores in regard to the STAI state and the trait scale, as well as the NEO-FFI Neuroticism and Openness scale. Likewise, in the case of the STAI state for DRD2 rs1076560 significant differences were found. Furthermore, while comparing the groups (e-cigarette users vs. controls) we noticed interactions for the NEO FFI Neuroticism and DRD2 rs1076560. The same was observed in the case of interactions significance while comparing groups (e-cigarette users vs. controls) for the STAI trait/scale and DRD2 rs1799732. Findings from this study demonstrate that psychological factors and genetic determinants should be analyzed simultaneously and comprehensively while considering groups of addicted patients. Since the use, and rapid increase in popularity, of electronic cigarettes has implications for public health, e-cigarette users should be studied holistically, especially younger groups of addicted and experimenting users.

## 1. Introduction

Currently, a growing popularity of electronic cigarettes (e-cigs) [1,2] may be observed, which may result in serious worries for the society of lawmakers, healthcare providers, and smokers [3]. E-cigs may be treated as a special tobacco product since they are smoke-free. The Internet is the main platform for their increasing sales (30–50%) [4]. Recently, the expected overall annual growth has reached 16.6%, which allows to estimate that the e-cig market will have reached $28 billion by 2022. The first generation of e-cigs was introduced in the European Union in 2006 and in the United States in 2007. Now, in the United States e-cigs are treated as a smoke-free tobacco/nicotine alternative used as an aid to smoke cessation and are presented as tobacco “harm reduction” products. Nonetheless, they are used more often as another source of nicotine, which may lead to the development and maintenance of nicotine dependence. Another important fact, the same like in the case of other abusive drugs, is that the rate and magnitude of the brain nicotine accumulation may significantly influence its acute reinforcing effects [5,6,7,8,9,10,11]. When the continuum of abuse liability in connection with nicotine-containing products is analyzed, it may be noticed that it is quite similar to the continuum of rapidity of nicotine distribution to the brain (which means that it is the highest for cigarettes and the lowest for nicotine patches). Moreover, the studies of experienced e-cig users emphasized that concentration of nicotine in venous blood was comparable to those observed after cigarette smoking [12,13,14] with a rapid delivery to the brain. That fact allows to conclude that a rapid brain uptake promotes a smoking reward. Therefore, e-cigarettes might support a degree of nicotine dependence and serve as non-combustible substitutes for cigarettes [15]. However, various addictions are emerging in our time, including behavioral addictions that pose a public health threat. That is why in our study we decided to analyze three DRD2 gene polymorphisms in connection with personality traits and anxiety, bearing in mind the fact that nicotine addiction is multi-factorial and multigenic, and that psychological factors may affect not only the addiction or its absence, but also determine the form and type of addiction.

In the literature we may find multiple examples of association between genetics and addiction. Motivational processes enhancing learning and exhibition of goal-directed behaviors are influenced by dopaminergic signaling—a system influencing substance use disorders. The evidence for it is discussed in the study concerning electrophysiological recordings from midbrain dopamine neurons in monkeys [16,17]. Numerous studies on animal models have analyzed striatal dopamine D2-type receptors as elements conditioning vulnerability and resilience to addiction [18,19,20,21,22,23,24,25]. They include these findings which say that the striatal D2-type receptor availability influences subsequent cocaine self-administration and that the striatal D2-type receptors and dopamine transporters show long-lasting neuroadaptations to stimulant administration in nonhuman primates [26,27]. Human studies also concluded that a below control striatal dopamine D2-type receptor availability in addicted individuals makes them prone to various substances such as stimulants, heroin, alcohol, and tobacco [28,29,30].

The dopamine receptor D2 gene (DRD2) is located in the chromosome 11q23 and spans 6556 kb. The DRD2 gene includes 8 exons that undergo transcription to mRNA of 2713 kb that is translated to protein of 443 AA. Skipping the sixth exon leads to production of a short form of receptor in opposition to a 29 AA longer form of receptor protein. The two isoforms of D2 receptors differ in their affinities for inhibitory G-proteins [31]. The D2 receptor is engaged in sending reward feelings connected with addictive substances. Polymorphic variants in the DRD2 gene and D2 receptor may be linked with substance use disorders, dependence and its endophenotypes [32,33,34,35]. The area of genetics of behavior and psychiatry is concentrated on the analysis of certain polymorphisms, including the DRD2 gene re1076550 located in intron 6. The occurrence of the T allele of rs1076560 results in a lower expression of the short isoform (D2S) in relation to the long one (D2L) in the caudate putamen and prefrontal cortex [36]. The other polymorphism analyzed in the context is rs1079597 (Taq1B) located in intron 1 of the DRD2 gene and it is noticeably associated with alcohol dependence [37], as well as other substance dependencies [38,39], including nicotine [40]. Another example of a genetic influence on nicotine dependence [41,42,43,44] is demonstrated by rs1799732 (−141C Ins/Del) located in the 5’ region of the DRD2 gene.

Following the reports from the literature, one could say that personality traits may become a predisposing addiction factor. One which is popularly described is impulsivity, which we define, according to Baratt, as “acting under the pressure of the moment”. This is a psychological characteristic which describs a patient, which may become the starting point for defining the endophenotype [45]. What is also popularly described is the so-called “novelty seeking” a characteristic quality of addicts, which shows strong correlations with another dopaminergic system polymorphism, i.e. DRD4, where a number of tandem repeats are connected by a functional polymorphism with the need for adventure seeking and linked with receptor sensitivity to dopamine uptake [46]. Therefore, research combining these features may enrich the search area at the level of basic research into addiction. Changes in the dopaminergic system may be a determinant of “search” in the case of addicts.

Behavior, lifestyle and maintenance of proper functions in a lifetime is influenced by personality. The Big Five [47,48,49], is a factor model which is most often used in personality research. It consists of five traits: Openness, Conscientiousness, Extraversion, Agreeableness and Neuroticism. The differences among people are conditioned by these traits and associated with behavior, emotions, motivation and cognition [50]. People with high neuroticism show a high tendency towards mood changes, and often experience feelings of anxiety, worrying, anger, fear, frustration, jealousy, guilt, envy, depressive moods and loneliness. [51,52]. Openness is a personality trait associated with intelligence and divergent thinking. It was found that openness depends on the function of dopamine, especially in the prefrontal cortex [53]. Conscientiousness is a quality defined as a tendency to control impulses and acts in a way which is socially acceptable [54]. Extraversion is characterized by sociability, assertiveness and excitability. Extraverted people may seem more dominant in the social environment, as opposed to people who are locked in it [55]. Agreeableness, however, is a tendency towards compassion and cooperation, and also includes attributes such as altruism, trust and other pro-social behaviors. Higher anxiety levels are associated with susceptibility to substance dependence. A number of studies describe the correlation of addiction with drug traits measured using the STAI questionnaire [56].

To analyze these personality traits the revised NEO personality inventory (NEO-FFI) is used most frequently [47]. The State-Trait Anxiety Inventory (STAI) is another tool that is used in addiction research. It allows measurement of both, the state—reaction to a particular situation, and the trait—general tendency to react in a certain way while being anxious [57]. As described above both genetics and personality are factors influencing addiction. Therefore, the aim of our current study was to evaluate the impact of both of these components on e-cigarettes usage, and also the impact of interaction of specific genetic variants with personality traits and anxiety. The analysis was performed by comparing three polymorphisms located in the DRD2 gene—rs1076560, rs1799732 and rs1079597 (Taq1B) and personality traits measured with the Big Five Questionnaire (NEO FFI), as well as anxiety measured with the State Trait Anxiety Inventory (STAI) in two groups of subjects—e-cigarette users and the non-smoking/ non-vaping control group. 

## 2. Materials and Methods 

### 2.1. Study Group and Control Group

The study was carried out in the Independent Laboratory of Health Promotion, Pomeranian Medical University in Szczecin, after receiving approval from the Bioethics Committee of the Pomeranian Medical University (KB-0012/106/16) and the informed, written consent of the subjects. The study was conducted on the group of 394 volunteers of Caucasian origin, including e-cigarette users exclusively (*n* = 144; mean age = 26.82, SD = 9.26, men constituted 52% and women 48%) and healthy controls (*n* = 250; mean age = 21.84 SD = 3.98, men constituted 61% and women 39%). The study group included volunteers who self-reported the use of e-cigarettes for at least two years, whereas the controls were grouped from healthy subjects, without a history of mental disorders, not using nicotine. The examined group was recruited in the West Pomeranian and Lubuskie Voivodships. The examined participants volunteered in response to advertisements posted on the Internet and in the form of posters and leaflets. Background differences between groups have not been adjusted for.

All subjects expressed their informed consent in writing to participate in the study. 

#### DNA Isolation and Genotyping

A standard procedure of collecting venous blood was applied to obtain genomic DNA that was used for genotyping in accordance with the real-time PCR method. Genotyping of rs1079597, rs1076560, rs1799732 in DRD2 gene was performed with the fluorescence resonance energy transfer in the LightCycler ® 480 II System (Roche Diagnostic, Basel, Switzerland) according to the standard manufacturer’s protocols.

### 2.2. Psychometric Tests

The following psychometric tests were also performed. The NEO Five-Factor Personality Inventory (NEO-FF) and the State-Trait Anxiety Inventory (STAI) questionnaires were applied for testing both the research group and the controls. The Personality Inventory (NEO Five-Factor Inventory, NEO-FFI) includes 6 components for each of the five traits—Neuroticism (Anxiety, Hostility, Depression, Self-consciousness, Impulsiveness, Vulnerability to stress), Extraversion (Warmth, Gregariousness, Assertiveness, Activity, Excitement Seeking, Positive Emotion), Openness to experience (Fantasy, Aesthetics, Feelings, Actions, Ideas, Values), Agreeableness (Trust, Straightforwardness, Altruism, Compliance, Modesty, Tendermindedness), Conscientiousness (Competence, Order, Dutifulness, Achievement striving, Self-discipline, Deliberation) [47]. The STAI measures anxiety as a state (A-state) including fear, discomfort, and arousal of the autonomic nervous system occurring temporarily in relation to a particular situation as well as the trait of anxiety (A-Trait), which may be described as a permanent and enduring disposition to experience stress, worries, and discomfort. To show the results of the NEO-FFI and STAI tests the sten scale was applied. We also analyzed the history of substance dependence and psychosis using the medical records and authors’ own survey. The Anxiety Inventory (STAI) questionnaire, is the tool frequently used for evaluating anxiety, a state that is mainly determined as a conditionally and transiently conditioned psychological state of an individual. What is more, anxiety may be treated as a relatively constant personality trait. The STAI questionnaire is based on two independent subscales, containing 20 statements each, which makes it more valuable in a personality traits assessment. One of them measures the anxiety trait X-1, whereas the other measures the anxiety trait X-2. Personality traits defined with the Big Five model (neuroticism, extraversion, openness, conscientiousness and agreeableness) are components which may be discussed on the basis of the NEO Five Factor Inventory questionnaire which includes 60 statements (self-reported) requiring from the participant to assess themselves with research approach. The results of the tests NEO-FFI and STAI were provided as sten scores. The conversion of raw scores into the sten scale was performed according to the Polish norms for adults, in which it was assumed that stens 1–2 meant very low scores; 3–4 meant low scores, 5–6 meant average scores; 7–8 were high scores, 9–10 were very high scores.

### 2.3. Statistical Analysis

Concordance between the genotype frequency distribution and the Hardy-Weinberg equilibrium (HWE) was tested with the HWE software (http://www.oege.org/software/hwe-mr-calc.html). The differences in frequencies of genotypes and alleles of the studied polymorphisms between e-cigarette users and control subjects were tested using the chi square test. The differences in mean scores of the NEO Five Factor Inventory traits (Neuroticism, Extraversion, Openness, Agreeability and Conscientiousness) and STAI (state and trait) between the analyzed groups were measured and compared using the U Mann-Whitney test. In order to analyze the interaction between genetic and personality features a multivariate ANOVA was performed. A multivariate analysis of factor effects ANOVA (NEO-FFI/ scale STAI/ × genetic feature × control and e-smokers subjects × (genetic feature × control and e-cig subjects)) was applied to analyze the relation between DRD2 variants, e-cigarette users and control subjects and the NEO Five Factor Inventory (NEO-FFI) and STAI. The condition of homogeneity of variance was satisfied (Levene test *p* > 0.05). A normal distribution was not observed in the case of analyzed variables. All computations were performed using STATISTICA 13 (Tibco Software Inc, Palo Alto, CA, USA) for Windows (Microsoft Corporation, Redmond, WA, USA).

## 3. Results

The observed DRD2 polymorphisms frequencies differed from expectations based on the Hardy-Weinberg theorem in the e-cigarette users and in the control group. There were no statistically significant differences between the e-cigarette users and the control group in the frequency of DRD2 rs1076560, rs1079597 and rs1799732 genotypes and alleles (Table 1). While comparing the controls and the case group subjects, we noticed significantly higher scores on the STAI state scale (M = 5.43 vs. M = 4.74, *p* < 0.0001), STAI trait scale (M = 5.82 vs. M = 5.20, *p* < 0.02), Neuroticism scale (M = 5.71 vs. M = 4.64, *p* < 0.01), Openness scale (M = 5.03 vs. M = 4.52, *p* < 0.01), (Table 2). 

Table 3, Table 4 and Table 5 present the results of our tests for interaction between genetic determinants and personality traits. These results are for 2 × 3 factorial ANOVA of the NEO Five-Factor Personality Inventory (NEO-FFI) and the State-Trait Anxiety Inventory (STAI) shown in sten scales. The STAI state scale compared for DRD2 rs1076560 (F_2388_ = 3.65 *p* < 0.05), which accounted for 1.8% of the variance, showed to be a significant result. Comparison of the groups (e-cigarette users vs. controls) for NEO FFI Neuroticism and DRD2 rs1076560 (Figure 1, F_2388_ = 4.11 *p* < 0.05), which accounted for 2.1% of the variance, demonstrated significant results for interaction. A similar situation was observed in the case of the STAI trait scale and DRD2 rs1799732 (Figure 2, F_2388_ = 4.66 (*p* < 0.05)), which accounted for 2.3% of the variance. Post-hoc analysis is shown in Table 3, Table 4 and Table 5.

In Figure 1 we observe that e-cigarette users who are carriers of the C/C and A/C genotypes show a higher level of neuroticism as measured by the NEO FFI test compared to the carriers of these genotypes in the non-smoking control group. In contrast, e-cigarette users who are carriers of the A/A genotype show a lower level of neuroticism compared to carriers of the A/A genotype in the control group. 

In Figure 2, we observe that e-cigarette users who are carriers of the ins/del and del/del genotypes show a higher level of anxiety measured by the STAI test compared to those carriers in the non-smoking control group.

## 4. Discussion

As demonstrated in the previous studies and in the deliberations cited in the Introduction, the dopaminergic system plays a role in nicotine addiction. However, there are scarce reports in the literature about e-cigarette users or other modern nicotine delivering systems. By examining the genetics of addiction, we have repeatedly proved the impact of selected DRD2 polymorphisms on addiction [58,59,60] whether in alcoholics or patients addicted to psychoactive substances. The first question that arises is whether it makes sense to study e-cigarette users as a separate group of addicts—will they develop a specific genotype (will it be the same as in other addictions)? One could say that only nicotine addiction as a genetic determinant should be studied. During the recruitment and examination of the patients, it was observed that e-cigarettes were chosen by young people, the majority of whom were under-age, particularly as a result of their easier accessibility and greater comfort of use (no unpleasant smell, conviction of lesser harmfulness, etc.) [61]. However, in our study we only show a group of young adults using electronic cigarettes. 

The psychometric tests are a reference and a variable that is an important element in the analysis. 

We have found that there is an interaction between the groups compared (e-cigarette users vs. controls) with DRD2 rs1076560 which affects the NEO FFI neuroticism score. Higher values on the NEO FFI neuroticism scale were obtained by e-cigarette users for C/C and A/C polymorphisms compared to the control group. In contrast, lower values were found for A/A polymorphism compared to the control group.

In scientific studies it has already been found that personality traits displayed by a patient are a clear factor that predisposes to addiction. Impulsivity should be mentioned first. Baratt describes this feature as “acting under pressure of the moment”, thus making prompt decisions. Thanks to the impulsivity analysis, it is possible to define subgroups of patients, and it may even become a start to isolate the endophenotype [45] It should be noted that the division of this characteristic is related to the motor quality and the quality of making choices. Choosing addictive substances (including illegal substances) and attempting to experiment with new substances or behaviors may lead to the development of addiction [62]. “Novelty seekers” are known in scientific reports as people with a greater predisposition to developing addiction. This search for new things (as a personality trait) may also be seen here in the context of looking for new addictive substances, new products on the market, as well as a desire to try something that is unknown and inaccessible. And here one should refer to the connection of this trait with the dopaminergic system. “Novelty seeking” correlates with the polymorphic variant of the DRD4 gene. Those who possess long alleles (over 5 VNTRs) in the functional polymorphism of that gene experience pleasure while looking for adventures, which, in turn, is associated with the receptor’s sensitivity to dopamine uptake [46]. In our study, e-cigarette users had higher values on the neuroticism scale for the C/C and A/C genotypes for the DRD2 rs1076560 gene polymorphism. 

The rs1076560 polymorphism was previously described as significant for the functioning of the dopaminergic system. Zhang also noticed that the SNPs rs1076560 and rs2283265, that are located in the fifth and sixth introns of DRD2, are significantly associated with decreased expression of D2S in comparison to D2L and D2 receptor density. The working memory and attention control test, based on an fMRI analysis, demonstrated a brain activity-modulating effect in correlation with variants in healthy participants [63]. Our previous study covered the element of openness to new experience in this context, in association with the ANKK1 Taq1A gene polymorphism. The increased effects of addicted subjects and control subjects were observed for the following parameters: STAI trait, Neuroticism scale, Extraversion scale and Agreeability scale. The highest effects for the ANKK1 Taq1A were observed for the STAI trait and they approximated to the statistical significance for the STAI state and Neuroticism [64].

What is also worth noting is the fact that the present study demonstrated correlations related to the results regarding anxiety. We found a significant effect on the STAI sten scale score for DRD2 rs1076560. We found that there was an interaction between the groups compared (e-cigarette users vs. controls) and DRD2 rs1799732 on the STAI trait scale. Higher values on the STAI trait scale were obtained by e-cigarette users for the ins/del and del/del polymorphisms compared to the control group. However, they did not differ for ins/ins polymorphism between e-cigarette users and the control group.

We found a relationship of anxiety symptoms lasting from early childhood with the influence on the availability of D2 in the striatum, which in turn is conditioned by the A1 allele of the ANKK1 gene [65].

A meta-analysis of 17 twin studies [66] showed a weighted mean heritability for nicotine dependence of 59% in male smokers and 46% in female smokers (average 56% for all smokers). Different analysis [67,68,69,70] resulted in a similar degree of heritability in smoking-related behaviors, including cessation and initiation. rs17999732 has been associated with many substance dependencies—nicotine [43], heroin [71] and opiates [72]. The above-mentioned polymorphism is located in the 5’ flanking region of the DRD2 gene. It was demonstrated that this region is extremely important in regulating gene expression [73,74]. However, the results of the studies in this regard are not uniform [75,76,77].

### Limitations and Future Directions

This study is not free from some limitations—it was conducted on the subjects of Caucasian origin, so we strongly feel that it should be repeated on subjects of different ethnicities. As we see the topic remains open. We still do not have an explicit answer to the question what determines e-smokers. On the basis of our study we can conclude positive initial results of correlation, however it is only a preliminary study. Therefore, we are cautious in drawing conclusions that would translate into a clinical image of the studied people. Translational research will take place when a larger group of people is studied. The group is still being recruited, subsequent genes are going to be examined, and in the future a GWAS study is going to be conducted. However, we should emphasize the fact, that psychological factors, simultaneously with a genetic component, may play a significant role in addiction.

## 5. Conclusions

We have not found statistically significant differences observed in two groups on genetic test alone. We noticed an interaction in the case of DRD2 rs1799732 polymorphism on the STAI trait scale. An increased level of anxiety was characteristic for the group of e-cigarette users with an ins/del genotype and del/del homozygotes in the control group. It may be related to the fact that coping with stress and negative emotional states are often motivational for addicted individuals to use drugs. Findings from this study demonstrate that psychological factors and genetic determinants should be analyzed simultaneously and comprehensively while considering groups of addicted patients. Since the use, and rapid increase in popularity, of electronic cigarettes has implications for public health, e-cigarette users should be studied holistically, especially younger groups of addicted and experimenting users. 

## Figures and Tables

**Figure 1 ijerph-17-00365-f001:**
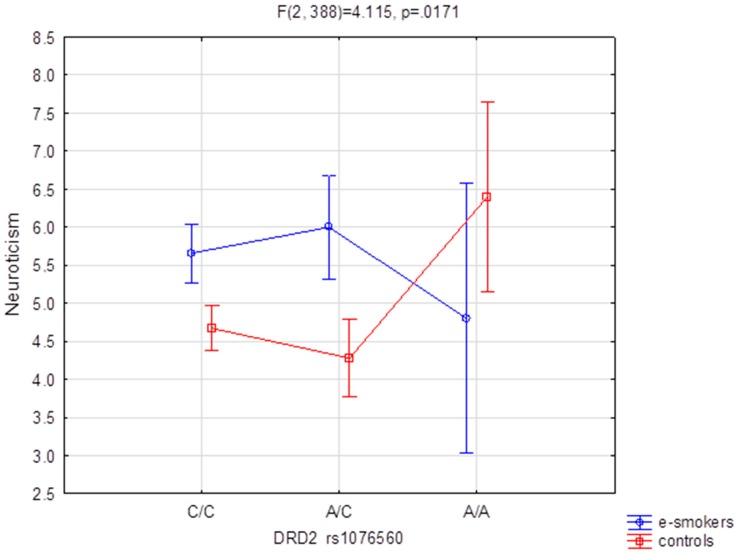
The group (e-cigarette users vs. controls) DRD2 rs1076560 polymorphism interaction for the NEO FFI Neuroticism.

**Figure 2 ijerph-17-00365-f002:**
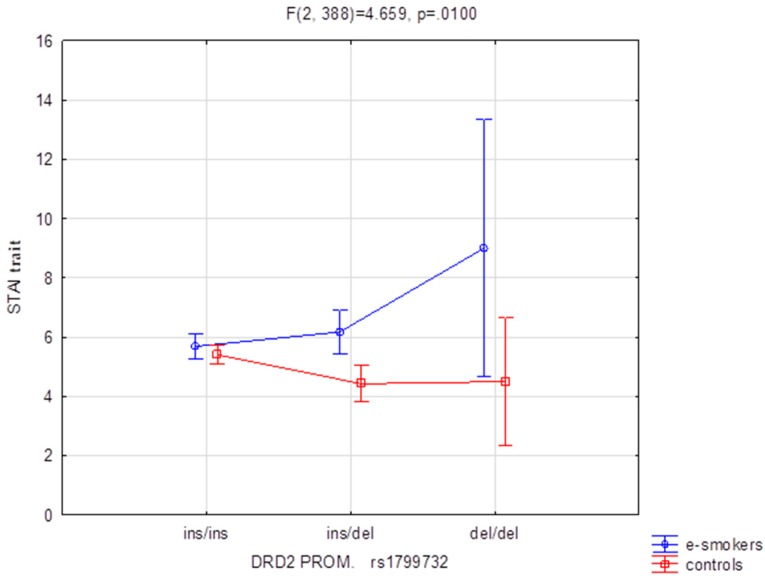
The group (e-cigarette users vs. controls) DRD2 rs1799732 polymorphism interaction for the STAI trait.

**Table 1 ijerph-17-00365-t001:** Frequency of genotypes and alleles of the DRD2 rs1076560, rs1079597 and rs1799732 polymorphisms in e-cigarette users and controls.

Genotypes/(Alleles)	E-Cig Users *N* = 144	Controls *N* = 250	χ^2^	*p* Value
***DRD2* rs1076560**				
C/C	105 (72.9%)	179 (71.6%)	0.112	>0.05
A/C	34 (23.6%)	61 (24.4%)		
A/A	5 (3.5%)	10 (4%)		
(C)	244 (84.7%)	419 (83.8%)	0.120	>0.05
(A)	44 (15.3%)	81 (16.2%)		
***DRD2* Tag1B rs1079597**				
G/G	104 (72.2%)	178 (71.2%)	0.089	>0.05
A/G	35 (24.3%)	62 (24.8%)		
A/A	5 (3.5%)	10 (4%)		
(G)	243 (84.4%)	418 (83.6%)	0.080	>0.05
(A)	45 (15.6%)	82 (16.4%)		
***DRD2* rs1799732**				
ins/ins	109 (75.7%)	197 (78.8%)	1.402	>0.05
ins/del	34 (23.6%)	49 (19.6%)		
del/del	1 (0.7%)	4 (1.6%)		
(ins)	252 (87.5%)	443 (88.6%)	0.210	>0.05
(del)	36 (12.5%)	57 (11.4%)		

*p*-statistical significance χ^2^ test, *N*—number of subjects.

**Table 2 ijerph-17-00365-t002:** STAI and NEO Five Factor Inventory results in controls and e-cigarette users.

STAI/NEO Five Factor Inventory/ (Sten Scale)	E-Cig Users (*N* = 144)	Control (*N* = 250)	Z (*p* Value)
STAI state/scale	**5.43 ± 2.35**	**4.74 ± 2.14**	**4.64 (<0.0001)**
STAI trait/scale	**5.82 ± 2.27**	**5.20 ± 2.21**	**2.51 (<0.02)**
Neuroticism/scale	**5.71 ± 2.03**	**4.64 ± 2.04**	**2.70 (<0.01)**
Extraversion/scale	6.14 ± 2.11	6.38 ± 1.91	−0.89 (>0.05)
Openness/scale	**5.03 ± 1.79**	**4.52 ± 1.55**	**2.83 (<0.01)**
Agreeability/scale	5.59 ± 2.29	5.52 ± 2.09	0.34 (>0.05)
Conscientiousness/scale	5.97 ± 2.20	6.23 ± 2.11	−1.32 (>0.05)

*p*-statistical significance U Mann’s test, N—number of subjects, M ± SD, Significances between-group differences are marked in bold print.

**Table 3 ijerph-17-00365-t003:** Differences in DRD2 rs1076560 and STAI/NEO Five Factor Inventory between control subjects and e-cigarette users.

STAI/NEO Five Factor Inventory (Sten Scale)			DRD2 rs1076560			Factor Effects ANOVA			
	E-Cig Users (*N* = 144)	Control (*N* = 250)	C/C (*N* = 284)	A/C (*N* = 95)	A/A (*N* = 15)	Factor	F (*p* Value)	ɳ^2^	Power (alfa = 0.05)
**STAI state/scale**	5.43 ± 2.35	4.74 ± 2.14	4.94 ± 2.19	4.87 ± 2.28	6.87 ± 2.23	intercept	**F_1388_ = 641.4 (*p* < 0.0001)**	**0.623**	**1.000**
						e-cig users/control	F_1388_ = 0.0001 (*p* > 0.05)	1 × 10^−8^	0.050
						**DRD2 rs1076560**	**F_2388_ = 3.65 (*p* < 0.05)**	**0.018**	**0.672**
						e-cig users/control × DRD2 rs1076560	F_2388_ = 1.81 (*p* > 0.05)	0.001	0.377
STAI trait /scale	5.82 ± 2.27	5.20 ± 2.21	5.42 ± 2.23	5.25 ± 2.25	6.73 ± 2.37	**intercept**	**F_1388_ = 676.3 (*p* < 0.0001)**	**0.635**	**1.000**
						e-cig users/control	F_1388_ = 0.12 (*p* > 0.05)	1 × 10^−6^	0.064
						DRD2 rs1076560	F_2388_ = 1.24 (*p* > 0.05)	0.006	0.271
						e-cig users/control × DRD2 rs1076560	F_2388_ = 2.48 (*p* < 0.10)	0.013	0.498
**NEO FFI** **Neuroticism/scale**	5.71 ± 2.03	4.64 ± 2.04	5.04 ± 2.09	4.89 ± 2.10	5.87 ± 2.10	**intercept**	**F_1388_ = 688.9 (*p* < 0.0001)**	**0.640**	**1.000**
						e-cig users/control	F_1388_ = 0.84 (*p* > 0.05)	0.002	0.149
						DRD2 rs1076560	F_2388_ = 0.31 (*p* > 0.05)	0.002	0.100
						**e-cig users/control × DRD2 rs1076560**	**F_2388_ = 4.11 (*p* < 0.05)**	**0.021**	**0.726**
NEO FFI Extraversion/scale	6.14 ± 2.11	6.38 ± 1.91	6.39 ± 1.95	6,15 ± 2.09	5.40 ± 1.76	**intercept**	**F_1388_ = 880.2 (*p* < 0.0001)**	**0.694**	**1.000**
						e-cig users/control	F_1388_ = 2.42 (*p* > 0.05)	0.006	0.341
						DRD2 rs1076560	F_2388_ = 2.76 (*p* < 0.10)	0.014	0.543
						e-cig users/control × DRD2 rs1076560	F_2388_ = 0.82 (*p* > 0.05)	0.004	0.191
NEO FFI Openness/scale	5.03 ± 1.79	4.52 ± 1.55	4.69 ± 1.66	4.78 ± 1.67	4.53 ± 1.64	**intercept**	**F_1388_ = 834 (*p* < 0.0001)**	**0.682**	**1.000**
						e-cig users/control	F_1388_ = 3.41 (*p* < 0.10)	0.009	0.453
						DRD2 rs1076560	F_2388_ = 0.05 (*p* > 0.05)	0.0002	0.057
						e-cig users/control × DRD2 rs1076560	F_2388_ = 0.46 (*p* > 0.05)	0.002	0.125
NEO FFI Agreeability/scale	5.59 ± 2.29	5.52 ± 2.09	5.61 ± 2.17	5.38 ± 2.17	5.47 ± 1.99	**intercept**	**F_1388_ = 646.5 (*p* < 0.0001)**	**0.625**	**1.000**
						e-cig users/control	F_1388_ = 0.34 (*p* > 0.05)	0.001	0.089
						DRD2 rs1076560	F_2388_ = 0.48 (*p* > 0.05)	0.002	0.128
						e-cig users/control × DRD2 rs1076560	F_2388_ = 0.29 (*p* > 0.05)	0.001	0.095
NEO FFI Conscientiousness/scale	5.97 ± 2.20	6.23 ± 2.11	6.08 ± 2.14	6.41 ± 2.07	5.07 ± 2.46	**intercept**	**F_1388_ = 765 (*p* < 0.0001)**	**0.663**	**1.000**
						e-cig users/control	F_1388_ = 0.51 (*p* > 0.05)	0.001	0.110
						DRD2 rs1076560	F_2388_ = 1.91 (*p* > 0.05)	0.010	0.397
						e-cig users/control × DRD2 rs1076560	F_2388_ = 1.20 (*p* > 0.05)	0.006	0.262

N—number of subjects, M ± SD. Significances between-group differences are marked in bold print.

**Table 4 ijerph-17-00365-t004:** Differences in DRD2 Tag1B rs1079597 and STAI/NEO Five Factor Inventory between control subjects and e-cigarette users.

STAI/NEO Five Factor Inventory (Sten Scale)			DRD2 Tag1B rs1079597			Factor Effects ANOVA			
	E-Cig Users (*N* = 144)	Control (*N* = 250)	G/G (*N* = 282)	A/G (*N* = 97)	A/A (*N* = 15)	Factor	F (*p* Value)	ɳ^2^	Power (alfa = 0.05)
STAI state/scale	5.43 ± 2.35	4.74 ± 2.14	4.96 ± 2.19	4.96 ± 2.30	5.87 ± 2.67	**intercept**	**F_1388_ = 575.8 (*p* < 0.0001)**	**0.597**	**1.000**
						e-cig users/control	F_1388_ = 1.06 (*p* > 0.05)	0.003	0.177
						DRD2 Tag1B rs1079597	F_2388_ = 0.96 (*p* > 0.05)	0.005	0.217
						e-cig users/control × DRD2 Tag1B rs1079597	F_2388_ = 0.52 (*p* > 0.05)	0.003	0.135
STAI trait /scale	5.82 ± 2.27	5.20 ± 2.21	5.44 ± 2.23	5.30 ± 2.24	6.07 ± 2.71	**intercept**	**F_1388_ = 632.3 (*p* < 0.0001)**	**0.620**	**1.000**
						e-cig users/control	F_1388_ = 0.14 (*p* > 0.05)	0.0004	0.066
						DRD2 Tag1B rs1079597	F_2388_ = 0.28 (*p* > 0.05)	0.001	0.094
						e-cig users/control × DRD2 Tag1B rs1079597	F_2388_ = 1.02 (*p* > 0.05)	0.005	0.228
NEO FFI Neuroticism/scale	5.71 ± 2.03	4.64 ± 2.04	5.05 ± 2.09	4.96 ± 2.11	5.13 ± 2.36	**intercept**	**F_1388_ = 637.9 (*p* < 0.0001)**	**0.622**	**1.000**
						e-cig users/control	F_1388_ = 2.99 (*p* < 0.10)	0.008	0.407
						DRD2 Tag1B rs1079597	F_2388_ = 0.03 (*p* > 0.05)	0.0001	0.054
						e-cig users/control × DRD2 Tag1B rs1079597	F_2388_ = 2.12 (*p* > 0.05)	0.011	0.434
NEO FFI Extraversion/scale	6.14 ± 2.11	6.38 ± 1.91	6.33 ± 1.95	6.22 ± 2.08	6.00 ± 2.14	**intercept**	**F_1388_ = 928.8 (*p* < 0.0001)**	**0.705**	**1.000**
						**e-cig users/control**	**F_1388_ = 5.52 (*p* < 0.05)**	**0.014**	**0.649**
						DRD2 Tag1B rs1079597	F_2388_ = 1.04 (*p* > 0.05)	0.005	0.232
						e-cig users/control × DRD2 Tag1B rs1079597	F_2388_ = 2.57 (*p* < 0.10)	0.013	0.512
NEO FFI Openness/scale	5.03 ± 1.79	4.52 ± 1.55	4.69 ± 1.64	4.78 ± 1.73	4.33 ± 1.63	**intercept**	**F_1388_ = 822.7 (*p* < 0.0001)**	**0.679**	**1.000**
						**e-cig users/control**	**F_1388_ = 4.95 (*p* < 0.05)**	**0.013**	**0.603**
						DRD2 Tag1B rs1079597	F_2388_ = 0.19 (*p* > 0.05)	0.001	0.080
						e-cig users/control × DRD2 Tag1B rs1079597	F_2388_ = 0.45 (*p* > 0.05)	0.002	0.123
NEO FFI Agreeability/scale	5.59 ± 2.29	5.52 ± 2.09	5.60 ± 2.15	5.33 ± 2.21	5.93 ± 2.05	**intercept**	**F_1388_ = 672.6 (*p* < 0.0001)**	**0.634**	**1.000**
						e-cig users/control	F_1388_ = 0.0003 (*p* > 0.05)	1 × 10^−6^	0.050
						DRD2 Tag1B rs1079597	F_2388_ = 1.00 (*p* > 0.05)	0.005	0.224
						e-cig users/control × DRD2 Tag1B rs1079597	F_2388_ = 0.31 (*p* > 0.05)	0.002	0.010
NEO FFI Conscientiousness/scale	5.97 ± 2.20	6.23 ± 2.11	6.06 ± 2.13	6.36 ± 2.09	5.87 ± 2.80	**intercept**	**F_1388_ = 802.5 (*p* < 0.0001)**	**0.674**	**1.000**
						e-cig users/control	F_1388_ = 0.02 (*p* > 0.05)	1 × 10^−5^	0.052
						DRD2 Tag1B rs1079597	F_2388_ = 0.92 (*p* > 0.05)	0.005	0.209
						e-cig users/control × DRD2 Tag1B rs1079597	F_2388_ = 0.26 (*p* > 0.05)	0.001	0.091

N—number of subjects, M ± SD. Significances between-group differences are marked in bold print.

**Table 5 ijerph-17-00365-t005:** Differences in DRD2 rs1799732 and STAI/NEO Five Factor Inventory between control subjects and e-cigarette users.

STAI/NEO Five Factor Inventory (Sten Scale)			DRD2 rs1799732			Factor Effects ANOVA			
	E-Cig Users (*N* = 144)	Control (*N* = 250)	ins/ins (*N* = 306)	ins/del (*N* = 83)	del/del (*N* = 5)	Factor	F (*p* Value)	ɳ^2^	Power (alfa = 0.05)
STAI state/scale	5.43 ± 2.35	4.74 ± 2.14	5.06 ± 2.26	4.77 ± 2.15	4.60 ± 2.30	**intercept**	**F_1388_ = 156.7 (*p* < 0.0001)**	**0.288**	**1.000**
						**e-cig users/control**	**F_1388_ = 5.56 (*p* < 0.05)**	**0.014**	**0.653**
						DRD2 rs1799732	F_2388_ = 0.61 (*p* > 0.05)	0.003	0.151
						e-cig users/control × DRD2 rs1799732	F_2388_ = 1.89 (*p* > 0.05)	0.010	0.393
**STAI trait** **/scale**	5.82 ± 2.27	5.20 ± 2.21	5.51 ± 2.21	5.14 ± 2.40	5.40 ± 2.61	**intercept**	**F_1388_ = 192.9 (*p* < 0.0001)**	**0.332**	**1.000**
						**e-cig users/control**	**F_1388_ = 6.64 (*p* < 0.01)**	**0.017**	**0.729**
						DRD2 rs1799732	F_2388_ = 0.89 (*p* > 0.05)	0.005	0.204
						**e-cig users/control × DRD2 rs1799732**	**F_2388_ = 4.66 (*p* < 0.05)**	**0.023**	**0.782**
NEO FFI Neuroticism/scale	5.71±2.03	4.64 ± 2.04	5.08 ± 2.09	4.93 ± 2.12	4.00 ± 2.45	**intercept**	**F_1388_ = 174.3 (*p* < 0.0001)**	**0.310**	**1.000**
						**e-cig users/control**	**F_1388_ = 7.00 (*p* < 0.01**	**0.0018**	**0.751**
						DRD2 rs1799732	F_2388_ = 0.16 (*p* > 0.05)	0.001	0.074
						e-cig users/control × DRD2 rs1799732	F_2388_ = 1.41 (*p* > 0.05)	0.007	0.302
NEO FFI Extraversion/scale	6.14 ± 2.11	6.38 ± 1.91	6.24 ± 2.02	6.46 ± 1.87	6.80 ± 2.17	**intercept**	**F_1388_ = 251.8 (*p* < 0.0001)**	**0.394**	**1.000**
						**e-cig users/control**	**F_1388_ = 5.57 (*p* < 0.05)**	**0.014**	**0.653**
						DRD2 rs1799732	F_2388_ = 0.62 (*p* > 0.05)	0.003	0.154
						e-cig users/control × DRD2 PROM. rs1799732	F_2388_ = 2.33 (*p* < 0.10)	0.011	0.471
NEO FFI Openness/scale	5.03 ± 1.79	4.52 ± 1.55	4.75 ± 1.60	4.58 ± 1.79	3.80 ± 2.77	**intercept**	**F_1388_ = 179.3 (*p* < 0.0001)**	**0.316**	**1.000**
						e-cig users/control	F_1388_ = 0.53 (*p* > 0.05)	0.001	0.113
						DRD2 rs1799732	F_2388_ = 2.29 (*p* > 0.05)	0.011	0.465
						e-cig users/control × DRD2 rs1799732	F_2388_ = 1.50 (*p* > 0.05)	0.008	0.319
NEO FFI Agreeability/scale	5.59 ± 2.29	5.52 ± 2.09	5.51 ± 2.15	5.54 ± 2.14	7.80 ± 2.28	**intercept**	**F_1388_ = 219.6 (*p* < 0.0001)**	**0.361**	**1.000**
						e-cig users/control	F_1388_ = 0.99 (*p* > 0.05)	0.005	0.221
						DRD2 rs1799732	F_2388_ = 0.36 (*p* > 0.05)	0.001	0.092
						e-cig users/control × DRD2 rs1799732	F_2388_ = 2.13 (*p* > 0.05)	0.011	0.436
NEO FFI Conscientiousness/scale	5.97 ± 2.20	6.23 ± 2.11	6.07 ± 2.17	6.22 ± 2.01	7.60 ± 2.70	**intercept**	**F_1388_ = 273.6 (*p* < 0.0001)**	**0.414**	**1.000**
						e-cig users/control	F_1388_ = 0.18 (*p* > 0.05)	0.0005	0.071
						DRD2 rs1799732	F_2388_ = 1.55 (*p* > 0.05)	0.008	0.328
						e-cig users/control × DRD2 rs1799732	F_2388_ = 0.53 (*p* > 0.05)	0.003	0.137

N—number of subjects, M ± SD. Significances between-group differences are marked in bold print.

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
