# Peer review of "Personality Traits or Genetic Determinants—Which Strongly Influences E-Cigarette Users?"

_ijerph, 2020, doi:10.3390/ijerph17010365_

Round 1

Reviewer 1 Report

This manuscript addresses an interesting topic related to e-cigarette use, namely, a possible linkage of genetic dopamine receptor polymorphism to personality traits.  The novelty of the subject matter was offset to a large degree by a number of concerns.

It is understandable that English is not the primary language of the authors, but there are a significant number of grammatical issues with the writing.  In addition, the implications of the findings, for treatment or otherwise, are not clear at all.  The authors should at least try and link their results to some aspect of tobacco treatment or public health.  Specific comments include:

Abstract:

DRD2 gene is not explained in the abstract – only in the intro.  Should give reader some brief explanation here. 

Line 40-42 – “… recommend comprehensively analyzing psychological and genetic factors in addiction” but do not give any explanation WHY this is important?  What are the clinical, treatment, or public health implications?

Line 42 – “e-smokers should be studies “holistically” “  - what does that mean?

“…especially in younger groups…” – why in younger groups?  Not explained.

Intro:

Line 49 – why are there worries among smokers?

Line 101-104 – the authors describe their methods (what the analyzed elements were), but do not state a “purpose” of the study?  What research questions were they trying to answer?  What was their hypothesis?  Otherwise, it appears that they were just “fishing” for a relationship between genetic results and personality results with no a priori plan. 

Methods:

Lines 110-112 should probably be in the result section

Line 119 – “sten scale” is not defined.

Line 122 – “ma” = typo?

Lines 141-146 – the authors are obviously genetic scientists.  However, for the readership, these lines are likely not very important and can be deleted to save space.

Results:

Lines 166-169 – these differences are statistically significant, but can the authors comment if they are clinically significant differences?  It would be useful if the authors in the Methods section reported the SCALES for these instruments?  Is it a 0-10 scale?  0-100?  See also for Table 3 – need to know the scales.

Tables 1 and 2 – seem to be complete, but only readers familiar with genetic results will understand them. 

Lines 182-184 – accounting for approximately 2% of the variance… is that a clinically important difference?

Discussion:

Lines 243-245 – if the addiction issues are due to nicotine, it should not mater if the drug is delivered by combusted tobacco or e-cigarette.    It would be useful if the authors stated whether their findings (linkage of polymorphisms to personality traits) are consistent with findings from cigarette smokers.

Overall:

Grammatical or typo examples can be found in the following lines:  25-28, 51-52, 64, 97, 122, 163 (“shortage of statistical significance”), Table 4 (*alpha).

Author Response

ANSWER

Dear Reviewer,

We would like to thank you for your valuable comments on the article. Below you will find our reply to your review. All changes are with a description or a comment and changes have been made to the manuscript (track changes in the tracking group on the review tab).

“Abstract: DRD2 gene is not explained in the abstract – only in the intro.  Should give reader some brief explanation here. “

Thank you for your comment. The description has been added in the Abstract.

“Line 40-42 – “… recommend comprehensively analyzing psychological and genetic factors in addiction” but do not give any explanation WHY this is important?  What are the clinical, treatment, or public health implications?”

Thank you for this comment. This explanation should be included in the manuscript - a description has been added.

Line 42 – “e-smokers should be studied “holistically” “  - what does that mean?

At this point, we wanted to consider addictions as a whole - as a group of symptoms and a series of causes. That is why in our study various psychological and genetic factors are combined. 

“…especially in younger groups…” – why in younger groups?  Not explained.

Thank you for addressing this issue. We noticed that trend when recruiting patients for the study. However, there is literature data that confirms this trend in the studied population - citation added (discussion 1 paragraph) .

Intro: Line 49 – why are there worries among smokers?

This statement is cited from other scientists’ studies. However, smokers' fears are understandable. Footnote added (1st paragraph in the Introduction).

Line 101-104 – the authors describe their methods (what the analyzed elements were), but do not state a “purpose” of the study?  What research questions were they trying to answer?  What was their hypothesis?  Otherwise, it appears that they were just “fishing” for a relationship between genetic results and personality results with no a priori plan. 

The purpose of the study has been specified in detail (last paragraph in the Introduction).

Methods: Lines 110-112 should probably be in the result section

The description of the study group was deliberately included in the Materials and Methods. However, we have added more detailed characteristics in this section, including the average age of both sexes.

Line 119 – “sten scale” is not defined

Added in the Materials and Methods.

Line 122 – “ma” = typo?

Thank you for this comment. Yes, it has been deleted.

Lines 141-146 – the authors are obviously genetic scientists. However, for the readership, these lines are likely not very important and can be deleted to save space.

We shortened the description.

Results: Lines 166-169 – these differences are statistically significant, but can the authors comment if they are clinically significant differences?  It would be useful if the authors in the Methods section reported the SCALES for these instruments?  Is it a 0-10 scale?  0-100?  See also for Table 3 – need to know the scales.

Scales added in the Material and Methods. We are unable to state that the results obtained are clinically significant, because it is only a dimension of someone’s personality. The patients, however, were all mentally healthy.

Tables 1 and 2 – seem to be complete, but only readers familiar with genetic results will understand them.

Thank you for this comment. Table 1 has been removed and the results kept in a form of a description. Table 2 - genotypes and alleles should not be left out as they show the distribution of genotypes and alleles, and this is very important in genetic analysis.

Lines 182-184 – accounting for approximately 2% of the variance… is that a clinically important difference?

We cannot answer this question at this stage. This is one of the first studies of this type, we do not yet have enough data which may be clinically correlated.

Discussion: Lines 243-245 – if the addiction issues are due to nicotine, it should not matter if the drug is delivered by combusted tobacco or e-cigarette.    It would be useful if the authors stated whether their findings (linkage of polymorphisms to personality traits) are consistent with findings from cigarette smokers.

The group of e-smokers is, of course, just one of the subgroups of nicotine users. In further publications we will introduce other subgroups - including traditional smokers and people using heated tobacco (iqos). However, it was important for us to examine this group separately with particular attention paid to personality traits.

Overall: Grammatical or typo examples can be found in the following lines:  25-28, 51-52, 64, 97, 122, 163 (“shortage of statistical significance”), Table 4 (*alpha).

All typos have been corrected.

Reviewer 2 Report

Comments and suggestions for authors provided in a separate Word document.

Author Response

ANSWER

Dear Reviewer,

We would like to thank you for your valuable comments on the article. Below you will find our reply to your review. All changes are with a description or a comment and changes have been made to the manuscript (track changes in the tracking group on the review tab).

I was expecting to see a Limitations section similar to most papers/studies. Sometimes papers don't have dedicated limitations sections, but still address the limitations within the discussion section, etc. I didn't see that when I read this paper. I would advise the authors to either include a specific Limitations section or add text to the discussion to address the limitations of this study.

Added “Limitations and future directions” in the Discussion.

Abstract

I think the abstract needs to be re-written. I don't think the current abstract is framed or set up well. It does not currently do a good job articulating the research question(s), approach, findings, and conclusions of the study (…)

The Abstract has been slightly reworded. Thank you for your comments regarding this issue. They proved to be very helpful and were used in the manuscript.

Introduction

The Discussion section does a good job of clearly stating that both genetics and personality traits have previously been shown to be linked with other types of addiction. The purpose of this study is to assess whether genetics and personality traits appear to be linked or associated with e-cigarette use. I would suggest moving some of that from the Discussion to the Introduction. I don't think the current introduction does a good job of clearly and succinctly highlighting (in general language) the large body of evidence supporting the link between genetics/personality traits and other types of addictions. If the Introduction does a better job setting that up, it will be easier and more logical to transition into laying out the research question(s) for this study.

Thank you for this comment. Particular fragments in successive sections have been filled in or moved from the Discussion to the Introduction.

Lines 67-92 deal with the background research and relevant context for the genetics portion of this study. I would suggest a simple intro for that content that clearly states (in plain language) that previous research has shown a strong link between genetics and addiction. That is done well and clearly in the Discussion section. I recommend doing that here so that the reader has that information upfront before reading this paper. That will provide the reader with important context to help them understand and interpret this study.

Thank you for this comment. Such a statement will greatly facilitate analysis of the content by the reader. Added [second paragraph in the Introduction].

Lines 93-100 deal with the personality traits component of this study. This text briefly acknowledges that different people have various personality traits and that certain tools or measures such as the NEO-FFI and the STAI are tools for assessing and categorizing personality that have been used in addiction research. A bit more information about these tools and measures here would be really helpful. Again, this reads pretty clearly to someone already familiar with the NEO-FFI and the STAI. But to someone not familiar with either, a little bit of text describing what type of information those tools/assessments provide (in just a bit more detail) would be really helpful for the reader at the beginning of the paper. Just some plain text to say that these tools can be used to assess the general level of anxiety that people experience and maybe also describe some of the NEO measures such as neuroticism and openness in plain language would be really helpful to readers.

Thank you for this comment - the description has been added in the Introduction (last paragraph) and descriptions included in the Materials and Methods section - in a new subsection entitled "psychometric tests".

Lines 101-104. This paragraph sets up the purpose and design of the current study. This is where the purpose of the study and the research question(s) it addresses should be clearly laid out. In its current state, this paragraph is insufficient to set up the paper or outline the purpose/objectives or research question(s) addressed by the paper. Again, I think it would be a place to say something along the lines of since it is known that genetics and personality traits are strongly associated with other types of addictions, this study aims to see whether similar associations exist for e-cigarette users. It would also be a place to say that specific genetic and personality comparisons were made between a group of e-cigarette users and a control group of non-users. I have re-read this paragraph several times. Each time I feel like something is missing or that the authors missed a chance to motivate or set up the study and paper. Revising this paragraph will improve the paper and make it easier for the reader to follow along with.

Thank you very much for this valuable comment. The goal has been clearly defined and added.

Materials and Methods: Study Group and Control Group

Lines 107-130. Split into two paragraphs. First paragraph is only a single sentence long. Editors have consistently advised me against 1-sentence paragraphs. I would recommend combining into a single paragraph.

Thank you for this comment - applied to the text.

Line 110. Mentions that the study group contains exclusive e-cigarette users. I interpret this to mean that they do not smoke cigarettes or use other forms of tobacco products. If that is correct, are these people who have NEVER smoked cigarettes and only use e-cigarettes, or are these people who are just not currently smoking cigarettes and only using e-cigarettes? Since many people have transitioned to e-cigarettes from traditional smoking or are dual users of both traditional cigarettes and e-cigarettes, I would like to ensure that I understand what type of individuals are in the study group.

As described in the Materials and Methods section, our study group consisted of e-cigarette users exclusively. They only used e-cigarettes during the study and two years prior to it.

Line 122. There appears to be a typo (some extra text here). Text says: "the state that is ma mainly". I believe the "ma" is extra text that needs to be deleted.

Thank you for this comment. That piece of text has been deleted.

There is no information here (or elsewhere in the paper) about the characteristics of the sample. I was expecting to see something about the race/ethnicity, sex, and age of the study participants. Not clear to me from reading the paper how similar or dissimilar participants were in terms of these characteristics. If there were variations in participants, I also didn't see anywhere in the paper whether the analyses controlled for those differences. Would recommend discussing this in the paper. I don't think it's necessary to include a summary table in the results, but some discussion of this would be helpful to the reader.

Thank you for this comment. We indeed overlooked that. An entry regarding ethnicity has been added (for average age and gender distribution).

I noticed that this subsection contains text describing both the genetic portion (lines 115-116) and then information on the personality traits measurements/tools/assessments (lines 116-130). But then on Lines 131-146 there is a specific subsection about the DNA isolation and genotyping. Would suggest shortening the description of the personality traits here and moving the description of those to their own subsection following the DNA Isolation and Genotyping. I would refer to it as something along the lines of "Personality Traits". In general, the paper tends to present the genetic stuff first and the personality traits stuff second. I would try to keep the order of presenting those topics consistent throughout the paper.

Thank you for this comment. A whole new section has been added - “psychometric tests” .

Materials and Methods: DNA Isolation and Genotyping

Lines 133-134: Another 1-sentence paragraph. Recommend combining this sentence with the next paragraph in this subsection (Lines 135-146).

Thank you for this comment. The two paragraphs have been combined.

Materials and Methods: Statistical Analysis

Right now, this section reads like a list of tests that were performed. It doesn't provide any context or sense of why each test was performed, what the outcome of those statistical analyses/tests tell you, and how those results relate back to the study questions/objectives. I don't think you need to add a lot here. But simple text like "To assess X, we performed such and such test". It's just not very clear how each of the different tests and analyses relate to your overall study design and purpose.

Thank you for this comment. The descriptions of statistical methods have been reformatted.

Line 154-156. This is one of the first times you mention the specific subcategories for the NEO Five Factor Inventory traits (neuroticism, extraversion, openness, agreeability, and conscientiousness). I believe that assessing each of the five factor inventory traits would be better mentioned and described earlier in the methods section along with your description of what data and personality measurements/assessments you used for this study.

Thank you for this comment. A whole new section has been added - “psychometric tests”.

As someone not familiar with the NEO inventory traits, it would be really useful for me to see a brief description or definition of what each of those things represents. For instance, what to neuroticism refer to? A simple definition will help the reader conceptualize those different aspects of personality that are being examined and considered in this study.

Thank you for this comment. The description has been added in the Introduction and in the Materials and Methods subsection.

Line 155. A comma is missing between the words "agreeability" and "conscientiousness".

Thank you for your comment and attention. Corrected by adding “and”.

Results

In general, I found the Results section to be consistently lacking in detail or information needed for me to interpret the results. Because I am not intimately familiar with genetics and the specific psychological measures and scales used to assess personality traits, the description in the Results section doesn't really help me understand the findings of the study. Some additional text to help the reader understand the results would be helpful. Specific comments are below. I think the authors rely on the results in the tables and figures too much without providing sufficient text to help the reader review, interpret, and understand the results. Without being intimately familiar with the measures, and the format of the analysis results/output presented in the data tables and figures, it is not immediately obvious or easy for a reader to understand what the results and findings are. Additional text to explain the findings, provide context around the results, with appropriate references to the tables and figures, would really improve this section.

Thank you for this comment. We conducted a team consultation to determine how to show our results more clearly. Table 1 regarding the HW law was deleted so that the reader wouldn’t have to be occupied with such details. However, the table descriptions must be written using a specialized language in order not to lose their meaning. We also tried to describe the results in the Discussion more clearly.

Lines 161-169. The text is currently three separate paragraphs. Each paragraph is only 1 sentence long. Recommend combining this text into a single paragraph.

            Thank you for this comment. The two paragraphs have been combined.

Lines 163-165. Text says "we observed a shortage of statistical differences between the e-cigarette users and the control group". Based on the p-values in Table 2, there were no statistically significant differences between the groups for any of the comparisons. Instead of saying a shortage was observed, I suggest that the authors instead say that there were no statistically significant differences.

Thank you for this comment. Changed as recommended.

Because the authors did not provide definitions or examples of the personality traits measures in the methods section, it is hard to meaningfully interpret the results presented in Table 3 without having prior familiarity with those measures.

Thank you for your comment – proper description added in the Introduction and in the Materials and Methods.

My previous comment is particularly the case for the STAI state scale and the STAI trait scale. It is not clear to me what each of those represent and how to interpret higher numbers. My understanding after reading the paper is that each of those is a measure of anxiety. As such, I would assume that higher values for those measures indicates higher anxiety. I would interpret the higher values for those two measures among the e-cigarette users to indicate that the e-cigarette users had significantly higher anxiety, based on those two measures. It would also be helpful for the authors to explain in the paper what is different between those two measures. If both the STAI state and STAI trait scales are measures of anxiety, what is different between them? And why did the authors include both in the study and the paper instead of just one? Knowing that information before reviewing the results presented in Table 3.

Thank you for your comment – proper description added in the Materials and Methods.

I have similar comments for each of those other NEO five factor measures. Without having neuroticism, extraversion, openness, agreeability, and conscientiousness defined or explained in the methods section, it is not clear what higher values for those measures represent. Extraversion and conscientiousness seem somewhat self-explanatory. But what does higher neuroticism mean? Also, what is openness with respect to — openness to what? I can see that the e-cigarette group had statistically higher neuroticism and openness. But without having a sense of what those scales are measuring, it is difficult to understand or interpret what the findings mean in a practical sense.

Thank you for your comment – proper description added in the Introduction and in the Materials and Methods.

Lines 180-187. This is the part of the results section where the authors discuss tests for interaction between the personality traits and the genetic measures. The results of those interaction tests are not really described or summarized here. Instead, they are described in the Discussion section. At this point in the paper, the reader is only presented with the results tables for those tests (Tables 4-6). Looking at those tables is confusing to me. I cannot clearly understand or determine whether there were statistically significant interaction effects. Some text describing which interaction effects were statistically significant would be both appropriate and helpful here at the end of the results section.

Thank you for this comment. The description of interaction is in both the results and the Discussion.  It may help the reader understand and analyze the results.

Figure 1 and Figure 2 graphically present results from some of the tests for interaction between the genetics and the personality traits. Each figure is referenced in the text of the results section. But the text doesn't really summarize what the figure is presenting, and how to interpret those results within the context of the analysis. Suggest adding additional text describing the information/analysis results that the figures are presenting.

Thank you for this comment. Additional descriptions have been placed below the figures.

Discussion

A lot of the information presented in the Discussion would work much better in the introduction to set up the study and prepare the reader. For example, in the first paragraph (lines 240-245) describe why they decided to analyze specific genetic and personality traits in this study. I would argue that that should be one of the first things presented in the introduction at the beginning of the paper.

Lines 250-257. This is more background information related to why personality traits are important, particularly with respect to addiction. I think this would also work much better in the introduction for the paper.

Thank you for this comment. The information has been moved to the Introduction.

Lines 270-272. These are results. Fine to repeat here in connection with other research. But I feel this is stated more clearly here than in the Results section. Should not feel like results are just appearing in the Discussion section.

The results are better described in the Results section, and here only the reminder description remains to make it easier for the reader to catch the context.

Lines 284-290. These results related to anxiety are described much more clearly here than in the results section. Would suggest having this type of description of the results in the results section. The discussion would be a place to reflect on the bigger themes emerging from the results. But the finding that there were significant differences in the groups with respect to some of the personality traits assessed (especially anxiety) and that there were some statistically significant interactions between personality traits and the genetic aspects considered is an important result that should have been made more clear in the Results section. Important to highlight in the discussion that anxiety appears to be an important consideration and that it may be even more important given the potential interaction between anxiety and the genetic characteristics.

The results are better described in the Results section, and here only the reminder description remains to make it easier for the reader to catch the context.

Lines 291-293. As with my previous comments. This is really important background information that I think would work better if presented early in the paper (particularly in the introduction). Stating early that higher anxiety levels have been previously shown to be associated with substance dependence is an important piece of relevant context and background information to have before viewing this study's results for anxiety. This text also links the use of the STAI questionnaire (specifically with respect to anxiety) and addiction in previous research. Definitely think that information would work better in the introduction to set up and motivate your study. Having that information at the beginning will provide readers with the context to interpret and make sense of your study results.

Thank you for this comment. The selected fragment has been moved to the Introduction.

Lines 304-309. As I mentioned in a previous comment, I was surprised to see that your paper did not include a Limitations section. I would think this would be a good place to include study limitations and potentially next steps. It's good to point out that this work is preliminary and future work may come from it.

One limitation worth noting might be the relatively small sample size as well as any limitations related to the characteristics of the sample. For example, if the study population was relatively homogenous, one might not expect to observe genetic differences. On the other hand, if the two groups were composed of different race/ethnicity, sex, and age groups, that might be expected to result in greater genetic differences. Would be interesting to compare e-cigarette users to non-users stratified by race/ethnicity, sex, and age. Would also be interesting to compare e-cigarette users across race, sex, and age to see if e-cigarette users of different races, sexes, and age groups have different genetic outcomes. Didn't see anything in the paper to address this. Ultimately, the question becomes whether or not the results from this study are generalizable to a larger population or not.

Thank you for this comment. Added "Limitations and future directions" section.

Conclusions

I agree with the conclusions and think the authors focused on the right things. But I think the text in this section could be rewritten to communicate the major conclusions in a more effective way. My understanding of the findings, based on reading the paper, is that the authors did not observe statistically significant differences across the groups with respect to the genetic measures. There were some statistically significant differences in some of the personality traits (as expected). However, tests for interactions between genetic traits and personality traits indicate some statistically significant interactions. Since the results suggest that there are important interactions, the authors are advising that both genetic and personality traits should be examined when studying e-cigarette users and that e-cigarette users should be researched and analyzed holistically since addiction is multi-factorial and driven by a number of determinants. That is essentially what the authors have said in the conclusion. But I think a revision and rewording of this section could make that point clearer and tell more of a story or "so what" for the study results.

Thank you for this comment. We are very cautious when it comes to discussing and drawing conclusions. The article has been complemented with the "Limitations and future directions” section, where our concerns are described. We hope that clearer conclusions will be reached when we examine a larger group of people, and, perhaps, more polymorphic loci of selected genes. For the time being we know that the area of research we pursue is right and that the assumptions of comprehensive analysis, including psychological traits and genetics, are well founded. However, we are still aware of our limitations - a relatively small study group and a fragment of the patients' genotypic characteristics.

Suggest making the Conclusion a single paragraph. Right now, it is three paragraphs, but each paragraph is only 1 sentence long.

Thank you for this comment. The amendment has been applied as recommended.

Reviewer 3 Report

Comments and suggestions:

Comparisons would be clearer if control subjects had never used nicotine. More information about subjects selection is needed. Background differences between groups have not been adjusted for. Results of interaction analysis have limited practical implications. Inadequate English presentation makes the manuscript difficult to follow.

END

Author Response

ANSWER

Dear Reviewer,

We would like to thank you for your valuable comments on the article. Below you will find our reply to your review. All changes are with a description or a comment and changes have been made to the manuscript (track changes in the tracking group on the review tab).

Comparisons would be clearer if control subjects had never used nicotine.

Yes, we know that - such a group is being recruited now. In the present study we have people who do not smoke nicotine - they are the control group. However, they are not "never smokers” only.

More information about subjects selection is needed.

Background differences between groups have not been adjusted for.

Thank you for this comment. The description has been added.

Results of interaction analysis have limited practical implications.

Thank you for this comment. We are very cautious when it comes to discussing and drawing conclusions. The article has been complemented with the "Limitations and future directions” section, where our concerns are described. We hope that clearer conclusions will be reached when we examine a larger group of people, and, perhaps, more polymorphic loci of selected genes. For the time being we know that the area of research we pursue is right and that the assumptions of comprehensive analysis, including psychological traits and genetics, are well founded. However, we are still aware of our limitations - a relatively small study group and a fragment of the patients' genotypic characteristics.

Inadequate English presentation makes the manuscript difficult to follow.

Thank you for this comment. The text has been proofread and corrected by a native speaker.

END

Thank you!

Round 2

Reviewer 1 Report

My previous comments have been addressed.

Author Response

Thank you.

Reviewer 2 Report

The revised version of this paper is substantially improved. Thank you to the authors for considering my comments and making efforts to address them. The authors have addressed all of my comments on the previous version of the paper.

I have a few additional comments on the revised paper. These are very minor comments/suggestions the authors may wish to consider. Please see the separate Word document that I prepared with additional feedback and comments (attached separately).

Author Response

Thank you for precious comments. They have been introduced to the text.

Reviewer 3 Report

Despite the modification, old issues remain. English presentation has room for improvement (e.g. long 3rd sentence of Abstract). Subject selection was not further described and background differences between groups were still not adjusted for.

Author Response

Thank you for the comments and for pointing to the important aspects.

The amendments have been introduced to the manuscript in the change tracking mode.

Added :

“The examined group was recruited in the West Pomeranian and Lubuskie Voivodships.

The examined participants volunteered in response to advertisements posted on the Internet and in the form of posters and leaflets. Background differences between groups have not been adjusted for.” (page 4, lines 145-149)

English was rechecked and the improvements have been introduced to the text.